# Effect of Tumor Burden on Tumor Aggressiveness and Immune Modulation in Prostate Cancer: Association with IL-6 Signaling

**DOI:** 10.3390/cancers11070992

**Published:** 2019-07-16

**Authors:** Chun-Te Wu, Yun-Ching Huang, Wen-Cheng Chen, Miao-Fen Chen

**Affiliations:** 1Department of Urology, Chang Gung Memorial Hospital at KeeLung, Keelung 20401, Taiwan; 2College of Medicine, Chang Gung University, Taoyuan 33302, Taiwan; 3Department of Urology, Chang Gung Memorial Hospital at Chiayi, Chiayi 61363, Taiwan; 4Department of Radiation Oncology, Chang Gung Memorial Hospital at Chiayi, Chiayi 61363, Taiwan

**Keywords:** prostate cancer, immune, tumor burden, CD44, IL-6

## Abstract

Local treatment is known to improve survival in men with locally advanced prostate cancer (LAPC), but the underlying mechanisms remain unclear. In the present study, we examined the role of tumor burden in tumor aggressiveness, as well as the pathway responsible for these changes. We used human and murine prostate cancer cell lines to examine the role of tumor burden in tumor aggressiveness, as well as its correlation with cancer stem cell (CSC) marker levels and IL-6 signaling. Furthermore, 167 prostate cancer biopsy specimens were analyzed in terms of correlations of IL-6 and CD44 levels with clinical patient characteristics. Data from preclinical models showed that larger tumor burden was associated with more aggressive tumor growth associated and increased CD44 expression. Using cellular experiments and orthotopic tumor models, we showed that CD44+ prostate cancer cells have CSC-like properties, enhanced epithelial–mesenchymal transition (EMT), and a more immunosuppressive microenvironment. There was a significant correlation between IL-6 and CD44 levels based on in vitro testing of clinical samples. Blockade of IL-6/STAT3 signaling attenuated the expression of CD44, CSC-like properties, and aggressive tumor behavior in vitro and in vivo. In conclusion, CD44 expression is significantly associated with tumor aggressiveness in prostate cancer and activation of IL-6 signaling leads to a suitable microenvironment for the induction of CD44 expression. Based on our study, reduced tumor burden was associated with attenuated IL-6 signaling and augmented tumor rejection in the microenvironment, which might mediate the benefit of clinical adoption with aggressive local therapy.

## 1. Introduction

Prostate cancer is a major health concern in male populations and one of the most common cancers worldwide [1]. The incidence of prostate cancer in the Asia-Pacific region is increasing [2]. The management of advanced prostate cancer continues to be a challenge, and there is no clear consensus regarding the optimal first-line treatment. The use of prostate-directed therapy for males with advanced prostate cancer has been assessed, and recent data support its addition to androgen deprivation therapy (ADT) [3,4,5]. The addition of aggressive local treatment provided a survival benefit for locally advanced prostate cancer (LAPC) patients versus those treated with conservative treatment. These results are supported by data from the STAMPEDE trial and several retrospective series [3,6,7,8]. However, the advantages of aggressive therapy with either surgery or radiotherapy (RT) for males with LAPC, and the underlying mechanisms thereof, require further investigation.

Cancer progresses in a stepwise fashion. Despite limited data regarding the role of aggressive local therapy in advanced prostate cancer, current studies suggest that local treatment decreases the tumor burden and increases the time required for castration-resistant prostate cancer (CRPC) to develop [9,10]. Tumor burden is one of the best predictors of long-term outcomes of prostate cancer [11,12,13]. Significant changes in whole-body tumor burden have been shown to predict PSA progression. Accordingly, we used an animal tumor model to explore whether tumor burden played a role in tumor progression, and the relationship of such progression with the tumor microenvironment, in support of the concept of local therapy for advanced prostate cancer.

Inflammation may play an important role in cancer progression [14,15]. Overexpression of IL-6 in many tumors has been associated with poor prognosis [16,17,18]. We previously reported that altered IL-6/STAT3 signaling is important for CRPC transition and aggressive behavior in prostate cancer [19]. Previous reports revealed a dominant role of IL-6 in the regulation and maintenance of different types of cancer-associated stem cells [20]. Cancer stem cells (CSCs) are believed to be a subset of tumor cells responsible for tumor initiation, growth, local invasion, and metastasis. CSCs are seen frequently in solid tumors, including prostate cancers [21,22,23]. CSC selection was also shown to have a major role in CRPC development [23,24]. Accordingly, we examined the relationships of CSC marker and IL-6 expression levels with tumor burden, to inform innovative future treatment approaches.

## 2. Materials and Methods

### 2.1. Tissue Specimens and Patient Characteristics

The Institutional Review Board of our Hospital approved this study (IRB No.: 201701798B0). A total of 167 patients with prostate cancer were enrolled and prostate needle core biopsy specimens collected from prostate cancer patients at diagnosis were subjected to immunochemical analyses. The clinical characteristics of the patients are shown in Table 1. Groups were compared using the Spearman-rank test. We also analyzed the predictive role of IL-6 and CD44 in biochemical failure, using data from 117 prostate cancer patients who were diagnosed before June 2014.

### 2.2. Immunohistochemical (IHC) Staining

Biopsies for each patient at diagnosis were subjected to IHC staining. The sections were incubated overnight with anti-IL-6 (1:20) and anti-CD44 (1:50) antibodies. The staining patterns on slides, containing 5–6 core biopsy specimens for each case, were assessed using Image Pro Plus 6.3 (IPP), combined with the semi-quantitative immunoreactive score (IRS). In the evaluation, all tumor cells on the slides were taken into consideration. The IRS was calculated by multiplying the staining intensity (graded as 0 = no staining, 1 = weak staining, 2 = moderate staining, and 3 = strong staining) by the percentage of positively stained cells (0 = no staining; 1, <10% of cells stained; 2 = 10–50% of cells stained; and 3 = >50% of cells stained). An IRS score ≥2 was considered positive (for IL-6 and CD44 expression).

### 2.3. Cell Culture and Reagents

A human androgen-sensitive prostate cancer cell line, LNCaP, and a transgenic adenocarcinoma of the mouse prostate (TRAMP) cancer cell line, TRAMP-C1, were cultured as described previously [19]. The TRAMP model is one of the most well-known mouse models of prostate cancer in the immunocompetent host. Stable IL-6-silenced cancer cells were generated by transfecting TRAMP-C1 cells with the IL-6 silencing vector and cultured in medium containing puromycin for 4 weeks. Mouse recombinant IL-6 and IL-6-neutralizing antibody were purchased from R&D Systems (Minneapolis, MN, USA), and IL-6 silencing vector and control vector were purchased from Santa Cruz Biotechnology (Santa Cruz, CA, USA). The effects of IL-6 signaling in vitro were assessed using cells incubated in drug-containing medium for specific times, following pre-incubation in the presence or absence of 60 ng/mL IL-6 or 5 µg/mL IL-6 Ab for 48 h.

### 2.4. Mouse Tumor Models (Ectopic and Orthotopic)

Eight-week-old male C57BL/6J and nude mice were used as the tumor implantation model, with the approval of the experimental animal committee of our hospital (Approval No.: 2017102601). Transfected TRAMP-C1 cells (1 × 10^6^ cells per implantation or as indicated, six animals per group at least) were implanted subcutaneously in the ectopic tumor implantation model, and intraoperatively into the lateral region of the prostate gland in the orthotopic tumor implantation model. The extent of orthotopic tumor invasion and tumor size were measured at 2 weeks after implantation, or at the indicated times. The effects of IL-6 stimulation on tumor growth were also investigated in vivo. In the treated group, an intraperitoneal injection of IL-6 (100 ng per mouse, three times per week) was administered 1 day before tumor implantation. To explore the role of tumor burden in tumor promotion, we performed local excision when ectopic tumors reached a size of 0.5 cm^3^. We collected cancer cells from xenograft tumors at 2 weeks after local excision (cells of smaller tumor burden; STB); tumors not subjected to local excision showed continued growth for 2 weeks (cells of larger tumor burden; LTB) (Figure 1).

### 2.5. Flow Cytometric Analyses of Myeloid-Derived Suppressor Cells (MDSCs)

The co-expression of myeloid cell lineage differentiation antigens Gr1 and CD11b characterize MDSCs in mice [25]. Therefore, we used a specific Gr1 antibody (clone RB6-BC5), which reacted with a common epitope on Ly-6G and Ly-6C, and an antibody specific for CD11b (clone M1/70; BD Pharmingen, San Jose, CA, USA) to define mouse MDSCs as CD11b+Gr1+.

### 2.6. Immunofluorescence (IF) Analyses of Tissue Specimens

Frozen tissue specimens were cut into 5–8 μm cryostat sections. Antibodies specific for CD11b and CD3 were used to define MDSCs and tumor-infiltrating lymphocytes (TILs), respectively. The sections were incubated overnight at 4 °C with antibodies against CD11b and CD3, washed three times with phosphate-buffered saline (PBS), and incubated for 1 h with fluorescein or Texas Red-conjugated secondary antibodies. The positive staining signals were assessed by microscope from ten random fields and semi-quantitated by MetaMorph software.

### 2.7. Statistical Analysis

Samples were analyzed using Student’s *t*-test. Data are presented as the mean ± standard error of the mean (SD). Each experiment was performed independently at least twice. A probability level of *p* < 0.05 was taken to indicate statistical significance, unless otherwise stated.

## 3. Results

### 3.1. Tumor Burden after Local Treatment Was Correlated with Tumor Aggressiveness

The addition of local therapy is believed to improve the prognosis for LAPC patients. Our preclinical data (Figure 2a) showed that a combination of local RT and ADT resulted in smaller tumors compared to ADT alone based on ectopic tumors in immunocompetent mice. Angiogenesis is one of the mechanisms that promote tumor progression, and CD31 mediated endothelial cell–cell interactions involved in angiogenesis. Figure 2b demonstrated that the addition of local RT was associated with decreased tumor cell proliferation and attenuated angiogenesis compared to ADT alone. We also implanted various amounts of tumor cells (1 × 10^5^, 1 × 10^7^, and 1 × 10^9^) to examine the correlation between tumor burden and tumor growth in mice. Based on data from ectopic and orthotopic models (Figure A1a,b in Appendix A), implantation of larger amounts of tumor cells was associated with more aggressive tumor growth and shorter latency to CRPC development in mice. Furthermore, following implantation of larger amounts of tumor cells, xenograft tumors showed increased expression of CD44 and more marked epithelial–mesenchymal transition (EMT) changes (Figure A1c). To further explore whether the tumor burden induced by local excision mediated alterations in the aggressive behavior of prostate cancers, we collected cancer cells from xenograft tumors with or without local excision (STB vs. LTB cells) (Figure 1). Thereafter, we orthotopically implanted 5 × 10^6^ STB or LTB cells into murine prostate. As shown in Figure 2c, implantation of LTB cells resulted in more rapid tumor growth, associated with higher rates of extra-prostate invasion. EMT is an important event affecting the invasiveness of various types of cancers and is characterized by an increase in Twist and vimentin expression [26]. The IHC data (Figure 2d) showed that the xenograft tumor from LTB cells exhibited increased expression of invasion-related factors and attenuated epithelial characteristics in vivo.

### 3.2. Role of CD44 in Tumor Aggressiveness

CD44 is a well-characterized marker associated with treatment resistance and aggressive tumor growth [27]. As shown in Figure 3a, CD44 levels were significantly increased in cancer cells of LTB compared to those of STB, based on fluorescence-activated cell sorting (FACS) analysis. Furthermore, IHC data on ectopic tumors (Figure A2a) confirmed that LTB prostate cancer exhibited higher levels of CD44. To examine the role of CD44 in tumor behavior, we isolated overexpressed CD44 subpopulations of cells (CD44+ cells) from murine prostate cancer cell lines using flow cytometry and cell sorting. EMT is a process that results in cancer cell migration, invasion, and metastasis [28]. Various biomarkers have been screened to demonstrate the EMT process, including the loss of E-cadherin, as well as the upregulation of matrix metalloproteinase, vimentin, and N-cadherin. As shown in Figure 3b,c, orthotopic implantation with CD44+ cancer cells was associated with larger prostate tumors and more marked EMT changes in both male and castrated mice compared to CD44− cancer cells. Moreover, in an experimental lung metastasis model (tail vein injection of tumor cells in vivo), the rate of metastatic nodules in the lungs was significantly higher in mice with CD44+ cells than CD44− cells (Figure 3d). Using cellular experiments, we showed that CD44+ cells had higher migration ability compared to CD44- cells (Figure A2b).

### 3.3. The Expression of CD44 Is Correlated with CSC and PD-L1 Expression

CD44 is a marker of CSCs; it is also known to be relevant to a resilient subpopulation of cells associated with increased tumorigenesis, and has an immunosuppressive phenotype [29,30]. CSCs are known to be enriched and maintained by sphere formation in serum-free medium, at low adherence levels [31]. Murine prostate cancer cells were cultured as spheres that could be dissociated during growth to generate subsequent passages of spheres, thus indicating that sphere-forming cells (SFCs) possess self-renewal capability (Figure A3). As shown in Figure 4a,b, SFCs increased CD44 expression compared to parental cells. At the molecular marker level, EMT is characterized by a loss of E-cadherin and increased expression of invasion-related factors [32]. Figure 4b show that CD44+ cancer cells exhibited increased invasion-related factors and fewer epithelial characteristics. Furthermore, CSCs contribute to tumor development and treatment resistance, partly by attenuating immune surveillance within the tumor microenvironment [30,33]. It has been postulated that the CSC niche utilizes the PD-1/PDL-1 axis to inhibit the immune response. Accordingly, we also compared the immunogenicity of CD44+ cells. As shown in Figure 4c, in vitro, increased PD-L1 levels were found in CD44+ cells and SFCs compared to CD44− cells. Moreover, Figure 4d,e demonstrated that CD44+ tumors were associated with increased MDSC recruitment and attenuated infiltration of CD3+ T cells in vivo.

### 3.4. Role of IL-6 Signaling in the Expression of CD44 in Prostate Cancer

Previously, we showed that the IL-6 level is significantly correlated with tumor aggressiveness and the transition of CRPC to prostate cancer. Figure 5a and Figure A4a illustrate there were higher IL-6 levels in mice bearing LTB tumors, associated with increased CD44 expression levels (Figure 5b). Activation of the IL-6/STAT3 pathway plays an important role in different types of cancer-associated stem cells [20,34,35]. To explore the impact of IL-6 on the expression of CD44 and the subsequent changes in prostate cancer, we investigated the role of IL-6 signaling in CD44 expression in prostate cancer cells. Figure 5c,d and Figure A4b show that IL-6/STAT3 signaling significantly increased the expression level of CD44 and is associated with increased expression of EMT in vitro and in vivo. In addition, activation of IL-6/STAT3 is known to induce immunosuppression in cancer by upregulating PD-1/PD-L1 [30,36,37]. As shown in Figure 5e–g, inhibition of IL-6 attenuated PD-L1 expression and MDSC recruitment in the tumors and spleen of tumor-bearing mice, and increased CD3+ TILs. Moreover, when IL-6/ STAT3 signaling was inhibited by STAT3 siRNA, the decreases in CD44 and EMT-related protein levels were comparable to those induced by IL-6-neutralizing antibody. Therefore, activating IL-6/STAT3 signaling plays an important role in the induction of CD44-positive prostate cancer.

### 3.5. Role of IL-6 in Patients with Prostate Cancer

Using IHC analysis, Figure 6a shows representative slides stained positively and negatively for IL-6 and CD44 in human cancer specimens at diagnosis. Of the 167 prostate cancer patients, 67 (40%) showed overexpression of IL-6. The positive staining for IL-6 was significantly associated with CD44 staining and the serum level of IL-6 (Figure 6a,b). As shown in Figure 6c and Table 1, the positive staining for IL-6 and serum level of IL-6 were significantly associated with a higher clinical stage, higher PSA level, and higher Gleason score. Furthermore, based on survival analysis, clinical stage and IL-6 level (but not CD44 level) had predictive power regarding biochemical failure in prostate cancer patients (Figure 6d). These findings suggest that IL-6 could be used to predict clinical stage and outcome in prostate cancer patients.

## 4. Discussion

Appropriate treatment is an important challenge in patients suffering from advanced prostate cancer. A growing number of retrospective analyses have reported improved survival among patients with metastatic prostate cancer or LAPC treated with prostatectomy or local RT [6,7,38]. Previous studies on local therapy for LAPC have increased our understanding of the role of the primary tumor. Based on the hypothesis that the primary prostatic tumor may play an important role in aggressiveness and metastatic expansion, we explored whether primary tumor burden is involved in changes in aggressive behavior in prostate cancer. First, we showed that the addition of RT to ADT resulted in smaller tumors, associated with decreased cell proliferation and angiogenesis compared to ADT alone. In addition, we subcutaneously implanted various amounts of tumor cells to examine tumor growth in ADT-treated mice. Our results showed that a larger tumor burden was associated with rapid tumor growth and shorter latency to CRPC development. Furthermore, we used orthotopic models to replicate the tumor microenvironment and cell interactions in vivo. EMT is a crucial biological event involved in cellular transformation and appears to be functionally relevant to the invasive characteristics of epithelial tumors. The data shows that cells derived from prostate tumors with local excision (STB) were associated with slower tumor growth, decreased tumor aggressiveness, and attenuation of EMT changes with decreased MMP-9 and vimentin compared to tumor cells derived from xenograft tumors without local therapy (LTB).

CSCs are commonly found in solid tumors including prostate cancer, and CSCs play a dominant role in CRPC development [23,24]. CSCs have been shown to be involved in tumor development, metastatic dissemination, and treatment resistance in numerous cancer models [39,40]. CD44 is considered a marker of CSCs from many organs, including the prostate [41]. Molecular studies showed that CD44+ prostate cancer cells possess stemness characteristics and retain specific intrinsic properties of progenitor cells [42]. Our preclinical data revealed a higher percentage of CD44-positive cells in xenografts in the context of a greater tumor burden. Accordingly, we examined the correlations of tumor burden, the induction of CD44+ cells, and CSC-like properties, in the context of the complex interplay between the primary tumor, the microenvironment, and the transition to more aggressive disease. Based on FACS and IHC analysis, we showed that LTB tumors had more CD44-positive cells than STB tumors. Furthermore, CD44-expressing prostate cancer cells expressed more marked EMT changes, associated with higher invasiveness. CSCs can be enriched by sphere formation in serum-free medium, at low adherence [31]. We showed that SFCs increased the expression of CD44 compared to parental cells. Moreover, CSCs contribute to tumor development and treatment resistance, partly by attenuating immune surveillance within the tumor microenvironment [33,43]. It has been postulated that the CSC niche supported by mesenchymal stem cells utilizes the PD-1/PDL-1 axis to suppress inflammation and inhibit the immune response. Therefore, we further examined the differential expression of PD-L1 between CD44+ and CD44− cell subpopulations. Our data showed that PD-L1 was increased in CD44+ cells. In addition, CD44+ tumor was associated with increased recruitment of MDSCs, but attenuated recruitment of TILs. These findings suggest that tumor burden plays a role in tumor progression, mediated by the induction of CSCs associated with the EMT and a more immunosuppressive tumor environment.

Several studies have shown that a major function of IL-6 is to regulate and maintain different types of cancer-associated stem cells [20,35]. Our previous data showed that IL-6 was significantly correlated with tumor aggressiveness and the transition to CRPC of prostate cancer [19]. Our in vivo data revealed higher IL-6 levels and greater activation of STAT3 were found in mice bearing LTB tumors, in association with increased CD44 expression. Furthermore, IHC data on 167 clinical prostate specimens demonstrated a positive correlation between CD44 and IL-6 levels in prostate tumor specimens. Accordingly, we further identified a role for IL-6 in CD44 expression, CSC-like properties, and the TME in prostate cancer. IL-6 enhanced the expression of CD44, which is associated with increased p-STAT3. When blocking IL-6 using antibody or STAT3 inhibition, the expression levels of CD44 and EMT-related proteins in prostate cancer cells were significantly attenuated both in vitro and in vivo. Previous studies have shown that IL-6 signaling induces dysfunctional immune system responses in the tumor microenvironment [17,37]. The IL-6/STAT3 pathway has been reported to increase PD-L1 expression [36]. In immunocompetent mouse models, we showed that blockade of IL-6 in prostate cancer cells significantly attenuated IL-6 signaling, decreased the expression of CD44, and attenuated the immunosuppressive tumor microenvironment. We further examined the correlation between the levels of IL-6 and clinical characteristics of prostate cancer patients. The positive staining of IL-6 and serum IL-6 levels were significantly associated with a higher clinical stage, higher PSA level, and higher Gleason score. Furthermore, in survival analysis, clinical stage and overexpressed IL-6 level, but not CD44 level, predicted biochemical failure in prostate cancer patients. The data demonstrated that enhanced expression of IL-6 was correlated with clinical tumor burden and the risk of biochemical failure.

## 5. Conclusions

Based on our results, IL-6 signaling mediated the disruption of the complex interplay between primary tumor, the microenvironment, and the transition to more aggressive disease in prostate cancer with or without effective local therapy. The expression of CD44 in prostate cancer is significantly associated with tumor aggressiveness, and IL-6 signaling leads to a suitable microenvironment for the induction of CD44 expression. Based on our study, IL-6 is a potential biomarker of aggressive prostate cancer. Objectively measured IL-6 expression levels in prostate cancer biopsy tissues could be of additional value with respect to determining the biological aggressiveness of prostate cancer and may be useful to guide treatment decisions.

## Figures and Tables

**Figure 1 cancers-11-00992-f001:**
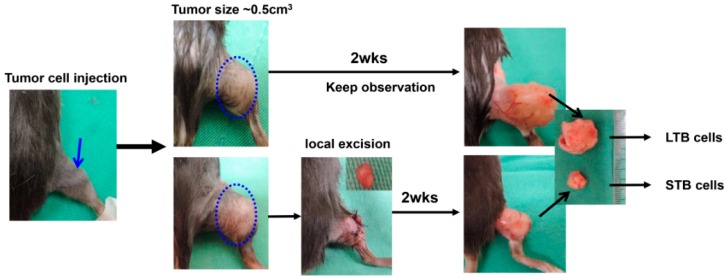
Tumor models for small tumor burden and large tumor burden. Transgenic adenocarcinoma of the mouse prostate (TRAMP)-C1 cells (1 × 10^6^ cells per implantation, or as indicated) were implanted subcutaneously. We performed local excision when ectopic tumors reached a size of 0.5 cm^3^. The cancer cells were collected from xenograft tumors at two weeks after local excision (cells of smaller tumor burden; STB); tumors not subjected to local excision and continued growth for two weeks (cells of larger tumor burden; LTB).

**Figure 2 cancers-11-00992-f002:**
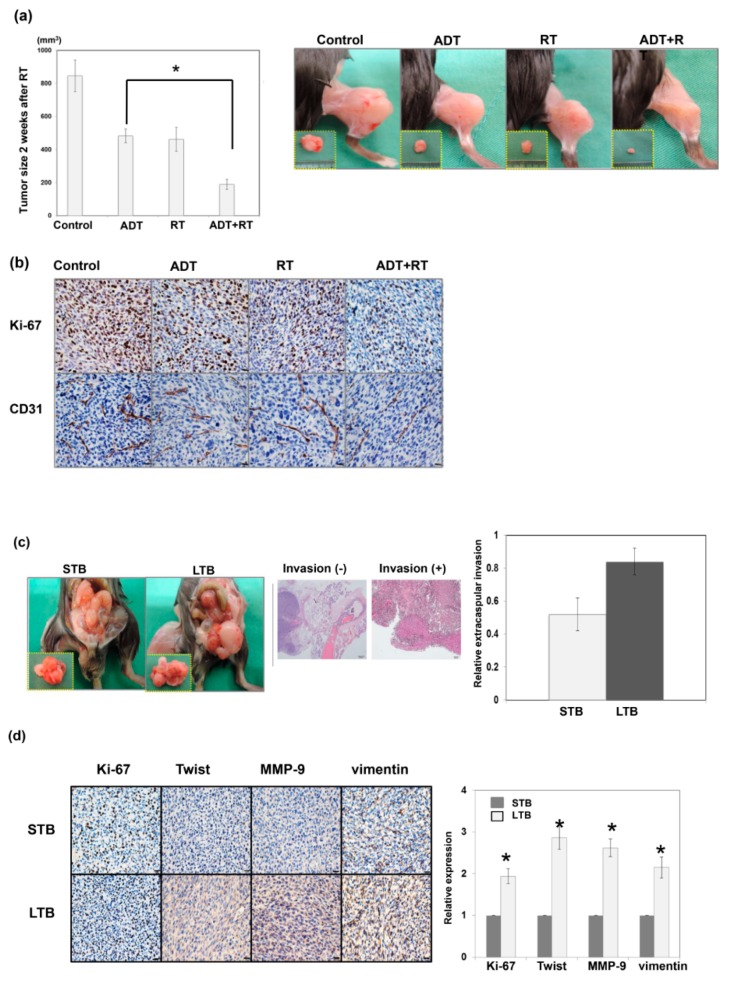
Effect of the addition of local RT combined with ADT on tumor control. (**a**) Effects of the addition of local RT on tumor growth were examined using of TRAMP-C1 ectopic tumor in mice with or without surgical ADT. The quantitative data are shown, and *Y*-axis represents the tumor size in mice two weeks after 15 Gy irradiation or sham irradiation. *, *p* < 0.05. Surgical androgen deprivation therapy (ADT), orchiectomy before TRAMP-C1 tumor implantation. The representative images were also shown two weeks after 15 Gy irradiation or sham irradiation. (**b**) The staining of ki-67 and CD31 prostate tumors at different treatment condition were evaluated by immunohistochemical (IHC) in vivo. Scale bars: 20 μm. (**c**) The effects of tumor burden on the invasive capacity of TRAMP-C1 prostate cancer in immunocompetent host were evaluated using orthotopic tumor implantation. The representative slides and quantitative data are shown. (**d**) The expressions on ki-67 and epithelial–mesenchymal transition (EMT)-related proteins in STB and LTB prostate tumors were evaluated by IHC in vivo. Scale bars: 20 μm. The *Y*-axis represents the relative ratio, normalized to the value of STB tumors. *, *p* < 0.05.

**Figure 3 cancers-11-00992-f003:**
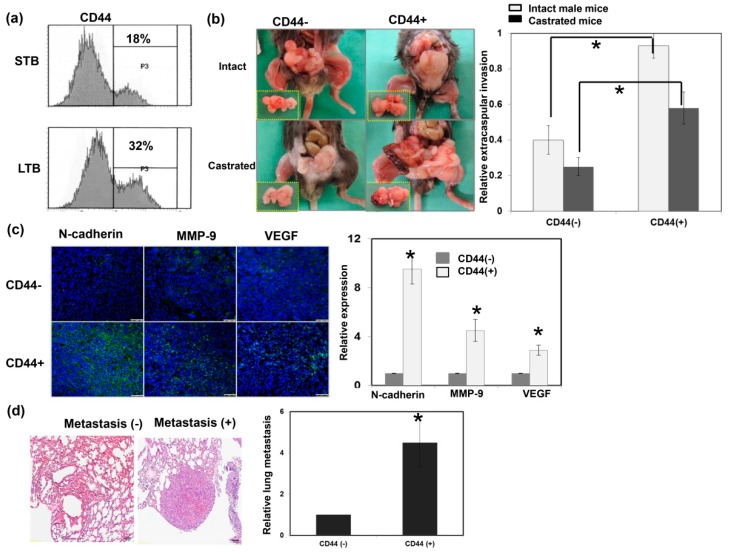
Role of CD44 in tumor aggressiveness. (**a**) The levels of CD44 were evaluated by fluorescence-activated cell sorting (FACS) for cancer cells of LTB compared to those of STB. (**b**) The invasive capacity was evaluated using orthotopic tumor model. The representative slides and quantitative data are shown. The *Y*-axis represents the ratio of mice presenting extracapsular extension normalized to that received orthotopic tumor implantation. Data points represent the means ± SEMs. *, *p* < 0.05. (**c**) Expressions of EMT-related proteins were examined by immunofluorescence (IF) in vitro (DAPI, blue; N-cadherin/VEGF/MMP-9, green). Scale bars: 50 μm. The quantification is to calculate the value of the cell number positive for targeted protein divided by the total cell number. The *Y*-axis represents the ratio normalized by the value of CD44 (−) cells. (**d**) Lung metastasis was determined by cells injected into mice via the tail vein and metastasis assayed. The representative slides and quantitative data are shown. *, *p* < 0.05.

**Figure 4 cancers-11-00992-f004:**
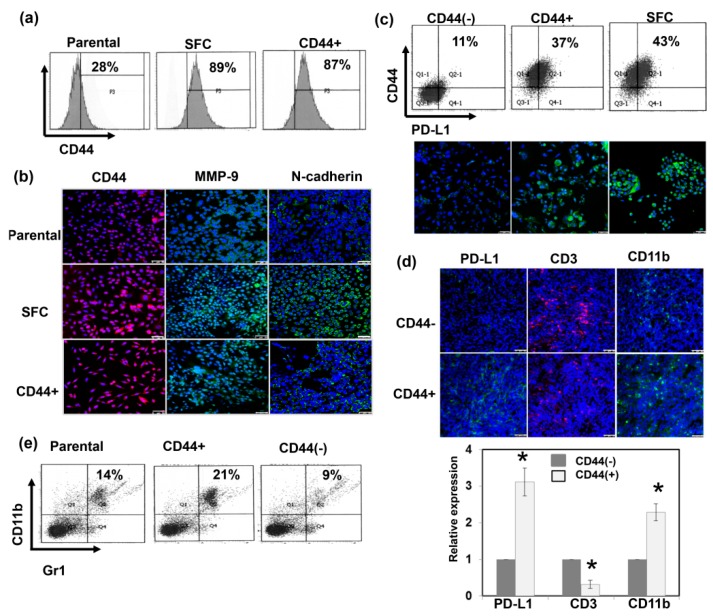
The expression of CD44 is correlated with CSC and PD-L1 expression. (**a**) The levels of CD44 were evaluated by FACS for sphere-forming cells (SFCs) and CD44+ cells compared with parental cells in vitro. (**b**) Expressions of EMT-related proteins were examined by IF in vitro. (DAPI, blue; N-cadherin/ MMP-9, green; CD44, red). Scale bars: 50 μm. (**c**) The levels of PD-L1 were evaluated for SFCs and CD44+ cells compared with CD44- cells by FACS and IF in vitro (DAPI, blue; PD-L1, green). Scale bars: 50 μm. CD44+ cancer cells were associated with increased myeloid-derived suppressor cells (MDSC) recruitment and attenuated infiltration of CD3+ T cells in tumors by (**d**) IF (DAPI, blue; CD3, Red; PD-L1/CD11b, green) with Scale bars: 50 μm, and (**e**) FACS (CD11b+/Gr1+ cells) in vivo. The quantification is to calculate the value of the cell number positive for targeted protein divided by the total cell number. The *Y*-axis represents the ratio normalized by the value of CD44 (−) cells. *, *p* < 0.05.

**Figure 5 cancers-11-00992-f005:**
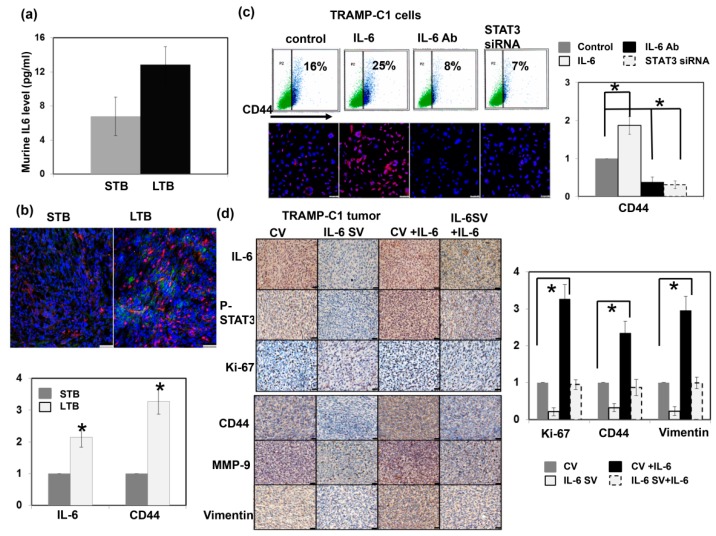
Role of IL-6 signaling in the expression of CD44. (**a**) IL-6 levels measured using ELISA in murine serum from STB and LTB tumors. *, *p* < 0.05 (**b**) The levels of IL-6 and CD44 were evaluated by IF for cancer cells of LTB compared to those of STB (DAPI, blue; IL-6, green; CD44, red). Scale bars: 50 μm. The quantification is to calculate the value of the cell number positive for targeted protein divided by the total cell number. The *Y*-axis represents the ratio normalized by the value of STB. *, *p* < 0.05. (**c**) The levels of CD44 were evaluated by FACS and IF (DAPI, blue; CD44, red) for TRAMP-C1 cancer cells with regulating IL-6 signaling in vitro (IL-6, cells stimulated with IL-6; IL-6 Ab, cells treated with IL-6 neutralization antibody; STAT3 siRNA, cells transfected with STAT3 siRNA). Scale bars: 50 μm. (**d**) The levels of CD44 and EMT changes were evaluated by IHC for TRAMP-C1 cancer cells with regulating IL-6 signaling in vivo (CV, cells transfected with control vector; IL-6 SV, cells transfected with IL-6 silencing vector; IL-6, mice stimulated with IL-6). Scale bars: 20 μm. The *Y*-axis represents the relative ratio, normalized to the value of CV tumors. *, *p* < 0.05. (**e**) The levels of PD-L1 and tumor-infiltrating CD3+ cells were evaluated by IF in vivo. Scale bars: 20 μm (**f**) The levels of MDSC (CD11b+Gr1+) recruitment in tumor and (**g**) The levels of tumor-infiltrating CD3+ cells in tumor and were evaluated by FACS.

**Figure 6 cancers-11-00992-f006:**
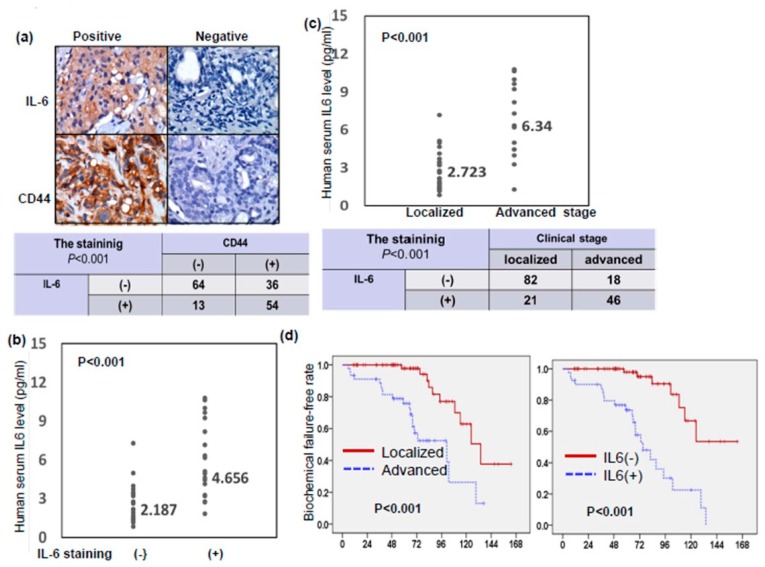
Role of IL-6 in patients with prostate cancer. (**a**) CD44 level was positively correlated with IL-6 expression in human prostate cancer specimens (*p* < 0.001). Representative slides of tumor specimens positively or negative staining for IL-6 and CD44 are shown. Scale bars: 60 μm. (**b**) IL-6 levels measured using ELISA in serum from cancer patients with IL-6 negative and positive staining. The value indicated the median. (**c**) IL-6 levels measured using ELISA in serum from cancer patients with clinical localized stage or advanced stage. (**d**) The differences of biochemical failure-free survival according to clinical stage and the staining of IL-6 in patients with prostate cancers.

**Table 1 cancers-11-00992-t001:** Clinical characteristic of patients.

Prostate Cancer IHC Staining	No. of Patients		*p* Value
	IL-6 (−)	IL-6 (+)	
Number	100	67	
Age			0.73
Median	72.3	73.1	
Range	48~83	47~85	
T stage			<0.001 *
T1–T2	83	23	
T3–T4	17	44	
Gleason score			<0.001 *
<7	54	14	
≥7	46	53	
LN and/or distant			<0.001 *
Negative	94	49	
Positive	6	18	
Pre-Tx PSA			<0.001 *
<20	66	19	
≥20	34	48	
NLR			
<3	73	15	<0.001 *
≥3	12	41	
Unknown	15	11	
Survival status			
Alive	90	55	0.11
Dead	10	12	

*: Statistically significant covariate.

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
