# Peer review of "Effect of Tumor Burden on Tumor Aggressiveness and Immune Modulation in Prostate Cancer: Association with IL-6 Signaling"

_cancers, 2019, doi:10.3390/cancers11070992_

Round 1
Reviewer 1 Report
The work entitled “Effect of tumor burden on tumor aggressiveness and 2 immune modulation in prostate cancer: Association with IL-6 signaling” by Wu et al. reported that the inhibition of IL-6/STAT3 signaling pathway was found to significantly downregulate the expression of CD44, EMT markers and PD-L1 in vitro and in vivo. The authors suggest that IL-6 could be used as a biomarker to predict the disease stage and clinical outcome in prostate cancer patients.
Below are some detailed comments that need to be addressed to improve the article.
Comments:
1. Fig. 2-6, Fig. S1-S3 – The quality of images is very poor and hence, it is hard to draw any conclusion based on these images.
2. The current data does not provide adequate data underlying EMT associated mechanisms. More EMT markers, such as N-cad, Vimentin, and SLUG should be added.
3. I don’t think there is a necessary to include Fig. 2a and b into the main figure. It would be interesting to look at the effect of the addition of local RT combined with ADT on both STB and LTB.
4. Fig 4. Images of spheroids should be shown. It would be also interesting to perform a sphere formation assay with STB and LTB tumours.
5. Fig 4c. Co-staining experiment between CD44 and PD-L1 should be performed.
6. Fig 4d. There is no appreciable differences in PD-L1 between CD44- and CD44+ cells.
7. Fig 5b. In line with my previous comment, co-staining experiment between IL6 and CD44 should be performed.
8. There are no images in Fig 5e, f and g.
Minor comments:
Please provide the number of eight-week-old male C57BL/6J and nude mice used for each experiments.
Fig. 2-6, Fig. S1-S3 - Please add scale bar and objective types to the images or figure legends.
Fig 3b, 4d, 5d, 6b,d, S2b and S3a – Label image - text overlaps image and Error Bar.
Fig 3b and d – Any statistical significance of these experiments?
Fig. 3 Line 178, Page 6 – Delete one of ‘was determined”.
Italic - in vitro and in vivo
Author Response
Response to reviewer 1
Thanks the reviewer’s comments. The changes that are in response to the Reviewers’ comments are shown in text with grey highlight and for reorganization are shown in blue text in the revised manuscript.
The work entitled “Effect of tumor burden on tumor aggressiveness and 2 immune modulation in prostate cancer: Association with IL-6 signaling” by Wu et al. reported that the inhibition of IL-6/STAT3 signaling pathway was found to significantly downregulate the expression of CD44, EMT markers and PD-L1 in vitro and in vivo. The authors suggest that IL-6 could be used as a biomarker to predict the disease stage and clinical outcome in prostate cancer patients.
Below are some detailed comments that need to be addressed to improve the article.
Comments:
1. Fig. 2-6, Fig. S1-S3 – The quality of images is very poor and hence, it is hard to draw any conclusion based on these images.
1) As mentioned by the reviewer, we modified the figures to be shown in higher resolution. Hope it could be better.
2. The current data does not provide adequate data underlying EMT associated mechanisms. More EMT markers, such as N-cad, Vimentin, and SLUG should be added.
2) As suggested by the reviewer, we modified the figures to show the data of vimentin in Figure 2d and Figure 5d, and the data of N-cadherin in Figure 3c, and Figure 4b. Hope it could provide adequate data underlying EMT.
We also added the paragraph “EMT is a process that results in cancer cell migration, invasion, and metastasis [28]. Various biomarkers have been screened to demonstrate the EMT process, including the loss of E-cadherin, as well as the upregulation of matrix metalloproteinase, vimentin and N-cadherin.” (Line 170), and added a new reference
[28] Zeisberg M, Neilson EG: Biomarkers for epithelial-mesenchymal transitions. J Clin Invest. 2009, 119:1429–1437
3. I don’t think there is a necessary to include Fig. 2a and b into the main figure. It would be interesting to look at the effect of the addition of local RT combined with ADT on both STB and LTB.
3) Thanks for the reviewer’s comments. Figure 2a & b showed that a combination of local RT and ADT resulted in smaller tumors compared to ADT alone, and Figure S1a & b demonstrated that a larger amount of tumor cells was associated with more aggressive tumor growth and shorter latency to CRPC development in mice.
We will further examine the effects of tumor burden on the treatment resistance in future.
4. Fig 4. Images of spheroids should be shown. It would be also interesting to perform a sphere formation assay with STB and LTB tumours.
4) As suggested by the reviewer, we added Figure S3 to demonstrate the image of spheroids compared to parental TRAMP-C1 cells in vitro.
5. Fig 4c. Co-staining experiment between CD44 and PD-L1 should be performed.
5) As suggested by the reviewer, we added the data of FACS using cells co-staining with CD44 and PD-L1 in Figure 4c. We also modified the IF data staining with PD-L1.
6. Fig 4d. There is no appreciable differences in PD-L1 between CD44- and CD44+ cells.
6) As suggested by the reviewer, we modified the IF data staining with PD-L1 in Figure 4d. Hope it could be better.
7. Fig 5b. In line with my previous comment, co-staining experiment between IL6 and CD44 should be performed.
7) As suggested by the reviewer, we added the data of IF using cells co-staining with CD44 and IL-6 in Figure 5b.
8. There are no images in Fig 5e, f and g.
8) Thanks for the reviewer’s mention. We added Figure 5e-g in the revised manuscript.
Minor comments:
Please provide the number of eight-week-old male C57BL/6J and nude mice used for each experiment.
1) As mentioned by the reviewer, we added “(1 × 106 cells per implantation or as indicated, six animals per group at least)” (Line 94)
Fig. 2-6, Fig. S1-S3 - Please add scale bar and objective types to the images or figure legends.
2) As suggested by the reviewer, we added the scale bar and represented the length of scale bar in Legend.
Fig 3b, 4d, 5d, 6b,d, S2b and S3a – Label image - text overlaps image and Error Bar.
3) We re-arranged the labels and images in Figures to avoid overlapping.
Fig 3b and d – Any statistical significance of these experiments?
4) As mentioned by the reviewer, we added marks “*” in Figure 3b and 3d, and “*, P <0.05.” in Legend to demonstrate the statistical significance
Fig. 3 Line 178, Page 6 – Delete one of ‘was determined”.
5) Thanks for the reviewer’s mention. We have deleted one “was determined” in the Legend of Figure 3d.
Italic - in vitro and in vivo
6) Thanks for the reviewer’s mention. We have modified “in vitro” and “in vivo” into “in vitro” and “in vivo”.
Reviewer 2 Report
In current study, Wu et al., investigated the tumor burden on tumor aggressiveness. In the orthotopic model, they showed that LTB cells are more invasive, which maybe contributed by elevated level of CD44. They further showed the positive correlation between CD44 and IL-6, suggesting IL-6 signaling as prediction marker for advanced PCa. Some of the findings are of interest, but a lot of the data lack a statistical analysis.
Some comments on figures.
1)Some of of IHC/ IF staining is hard to appreciate the difference, so it is better to have a quantified data representation.
2)In figure 3, the uncastrated condition can be called "intact" instead "male"
3)Is not clear why CD31 staining was done
4)what happens to the TRAMP tumors proliferation after exposing to IL-6 or IL-6 SV.
5)Is there any significance in Fig5a,5c, and the efficiency of STAT3 knockdown needs to be shown.
Author Response
Response to Reviewer #2:
Thanks the reviewer’s comments. The changes that are in response to the Reviewer’s comments are shown in text with grey highlight and for reorganization are shown in blue text in the revised manuscript.
In current study, Wu et al., investigated the tumor burden on tumor aggressiveness. In the orthotopic model, they showed that LTB cells are more invasive, which maybe contributed by elevated level of CD44. They further showed the positive correlation between CD44 and IL-6, suggesting IL-6 signaling as prediction marker for advanced PCa. Some of the findings are of interest, but a lot of the data lack a statistical analysis.
Some comments on figures.
1)Some of of IHC/ IF staining is hard to appreciate the difference, so it is better to have a quantified data representation.
1) Thanks for the reviewer suggestion. We added the quantitative data in Figure 2d, Figure 3c, and Figure 4d, Figure 5b, 5c and Figure 5d, and added the description in Legend. We also added “The staining patterns on slides, containing 5–6 core biopsy specimens for each case, were assessed using Image Pro Plus 6.3 (IPP), combined with the semi-quantitative immunoreactive score (IRS)” (Line 72), and “The positive staining signals were assessed by microscope from ten random fields and semi-quantitated by MetaMorph software.” (Line 121)
2)In figure 3, the uncastrated condition can be called "intact" instead "male"
2) As suggested by the reviewer, we changed “ male” into “ intact” in Figure 3b
3)Is not clear why CD31 staining was done
3) As mentioned by the reviewer, we added the description” Angiogenesis is one of the mechanisms that promote tumor progression, and CD31 mediated endothelial cell-cell interactions involved in angiogenesis. Figure 2b demonstrated that the addition of local RT was associated with decreased tumor cell proliferation and attenuated angiogenesis compared to ADT alone.” (Line 131) to describe why CD31 staining was done in Figure 2b. We also changed the data of CD31 staining into the data of vimentin staining in Figure 2d to demonstrate the EMT changes.
4)what happens to the TRAMP tumors proliferation after exposing to IL-6 or IL-6 SV.
4) As suggested by the reviewer, we added the data of ki-67 staining for TRAMP tumors after exposing to IL-6 or IL-6 SV. Expression levels of Ki-67 were evaluated by IHC analysis. Representative images and quantitative data are shown in Figure 5d.
5)Is there any significance in Fig5a,5c, and the efficiency of STAT3 knockdown needs to be shown.
5) As mention by the reviewer, we added the quantitative data for Figure 5c, and added marks“*” in Figure 5a and 5c, and “*, P<0.05.< span="">” in Legend to demonstrate the statistical significance. We also demonstrated that STAT3 siRNA significantly decreased the level of activated STAT3 in Figure S4b
Reviewer 3 Report
I have read with great interest the manuscript: Effect of tumor burden on tumor aggressiveness and immune modulation in prostate cancer: Association with IL-6 signaling. The study is interesting and conducted very thoroughly. Your study is a further evidence suggesting that IL‐6 plays a major role in the transition from hormone‐dependent to castration-resistant prostate cancer. You have analysed the addiction of RT to ADT in mice (Fig. 2), it should be very important also the correletion with IL-6. Because IL-6 is the topico f your study I suggest a deeper correlation of IL-6 in all the experiments . You have shown “increased expression of CD44 and more marked epithelial-mesenchymal transition (EMT) changes”, regarding this I suppose that EMT should be described in more detail, es the expression of N‐cadherin (marker of mesenchymal phenotype), and vimentin (cytoskeletal protein expressed by mesenchymal cells), should be evaluated in addition to E cadherin. Please revise the text for English language.
Author Response
Response to reviewer 3
Thanks the reviewer’s comments. The changes that are in response to the Reviewer’s comments are shown in text with grey highlight and for reorganization are shown in blue text in the revised manuscript.
I have read with great interest the manuscript: Effect of tumor burden on tumor aggressiveness and immune modulation in prostate cancer: Association with IL-6 signaling. The study is interesting and conducted very thoroughly. Your study is a further evidence suggesting that IL‐6 plays a major role in the transition from hormone‐dependent to castration-resistant prostate cancer. You have analysed the addiction of RT to ADT in mice (Fig. 2), it should be very important also the correletion with IL-6. Because IL-6 is the topico f your study I suggest a deeper correlation of IL-6 in all the experiments. You have shown “increased expression of CD44 and more marked epithelial-mesenchymal transition (EMT) changes”, regarding this I suppose that EMT should be described in more detail, es the expression of N‐cadherin (marker of mesenchymal phenotype), and vimentin (cytoskeletal protein expressed by mesenchymal cells), should be evaluated in addition to E cadherin.
1) Thanks for the reviewer suggestion. It is important regarding the correlation between IL-6 and local RT with or without ADT. I will keep it in mind for our study in future.
2) As suggested by the reviewer, we modified the figures to show the data of vimentin in Figure 2d and Figure 5d, and the data of N-cadherin in Figure 3c, and Figure 4b. Hope it could provide adequate data underlying EMT. We also added the paragraph “EMT is a process that results in cancer cell migration, invasion, and metastasis [28]. Various biomarkers have been screened to demonstrate the EMT process, including the loss of E-cadherin, as well as the upregulation of matrix metalloproteinase, vimentin and N-cadherin.” (Line 170), and “EMT is a crucial biological event involved in cellular transformation, and appears to be functionally relevant to the invasive characteristics of epithelial tumors.” (Line 283) We also added a new reference [28] Zeisberg M, Neilson EG: Biomarkers for epithelial-mesenchymal transitions. J Clin Invest. 2009, 119:1429–1437
Please revise the text for English language.
3) Thanks for the reviewer suggestion. The English in this document has been checked by a native speaker of English.

Round 2
Reviewer 1 Report
The manuscript has been significantly improved but the quality of some images still remains poor.
Author Response
Thanks the reviewer’s comments.
As mentioned by the reviewer, we modified the figures (including Figure 1-6 and Figure S1-4) to be shown in higher resolution (400dpi). Hope it could be better. We highly appreciate your precious comments which help to make this paper better quality.
Reviewer 3 Report
you have satisfied my concerns, the manuscript is now acceptable
Author Response
Thanks the reviewer’s comments. We highly appreciate your precious comments which help to make this paper better quality